# PROJECTED CANONICAL DECOMPOSITION FOR KNOWLEDGE BASE COMPLETION

## ABSTRACT

The leading approaches to tensor completion and link prediction are based on the canonical polyadic (CP) decomposition of tensors. While these approaches were originally motivated by low rank approximations, the best performances are usually obtained for ranks as high as permitted by computation constraints. For large scale factorization problems where the factor dimensions have to be kept small, the performances of these approaches tend to drop drastically. The other main tensor factorization model, Tucker decomposition, is more flexible than CP for fixed factor dimensions, so we expect Tucker-based approaches to yield better performance under strong constraints on the number of parameters. However, as we show in this paper through experiments on standard benchmarks of link prediction in knowledge bases, ComplEx, (Trouillon et al., 2016), a variant of CP, achieves similar performances to recent approaches based on Tucker decomposition on *all* operating points in terms of number of parameters. In a control experiment, we show that one problem in the practical application of Tucker decomposition to large-scale tensor completion comes from the adaptive optimization algorithms based on diagonal rescaling, such as Adagrad. We present a new algorithm for a constrained version of Tucker which implicitly applies Adagrad to a CP-based model with an additional projection of the embeddings onto a fixed lower dimensional subspace. The resulting Tucker-style extension of ComplEx obtains similar best performances as ComplEx, with substantial gains on some datasets under constraints on the number of parameters.

## 1 INTRODUCTION

The problems of representation learning and link prediction in multi-relational data can be formulated as a binary tensor completion problem, where the tensor is obtained by stacking the adjacency matrices of every relations between entities. This tensor can then be intrepreted as a "knowledge base", and contains triples (subject, predicate, object) representing facts about the world. Link prediction in knowledge bases aims at automatically discovering missing facts (Bordes et al., 2011; Nickel et al., 2011; Bordes et al., 2013; Nickel et al., 2016a; Nguyen, 2017).

State of the art methods use the canonical polyadic (CP) decomposition of tensors (Hitchcock, 1927) or variants of it (Trouillon et al., 2016; Kazemi & Poole, 2018; Lacroix et al., 2018). While initially motivated by low-rank assumptions on the underlying ground-truth tensor, the best performances are obtained by setting the rank as high as permitted by computational constraints, using tensor norms for regularization (Lacroix et al., 2018). However, for large scale data where computational or memory constraints require ranks to be low (Lerer et al., 2019), performances drop drastically.

Tucker decomposition is another multilinear model which allows richer interactions between entities and predicate vectors. A special case of Tucker decomposition is RESCAL (Nickel et al., 2011), in which the relations are represented by matrices and entities factors are shared for subjects and objects. However, an evaluation of this model in Nickel et al. (2016b) shows that RESCAL lags behind other methods on several benchmarks of interest. Recent work have obtained more competitive results with similar models (Balažević et al., 2019b; Wang et al., 2019), using different regularizers or deep learning heuristics such as dropout and label smoothing. Despite these recent efforts, learning Tucker decompositions remains mostly unresolved. Wang et al. (2019) does not achieve state of the art results on standard benchmarks, and we show (see Figure 3) that the performances reported by

Balažević et al. (2019b) are actually matched by ComplEx (Trouillon et al., 2016; Lacroix et al., 2018) optimized with Adam, which has less hyperparameters.

In this work, we overcome some of the difficulties associated with learning a Tucker model for knowledge base completion. Balažević et al. (2019b) use deep-learning mechanisms such as batch normalization (Ioffe & Szegedy, 2015), dropout (Srivastava et al., 2014) or learning-rate annealing to address both regularization and optimization issues. Our approach is different: We factorize the core tensor of the Tucker decomposition with CP to obtain a formulation which is closer to CP and better understand what difficulties appear. This yields a simple approach, which has a single regularization hyperparameter to tune for a fixed model specification.

The main novelty of our approach is a more careful application of adaptive gradient techniques. State-of-the-art methods for tensor completion use optimization algorithms with adaptive diagonal rescaling such as Adagrad (Duchi et al., 2011) or Adam (Kingma & Ba, 2014). Through control experiments in which our model is equivalent to CP up to a fixed rotation of the embeddings, we show that one of the difficulties in training Tucker-style decompositions can be attributed to the lack of invariance to rotation of the diagonal rescaling. Focusing on Adagrad, we propose a different update rule that is equivalent to implicitely applying Adagrad to a CP model with a projection of the embedding to a lower dimensional subspace.

Combining the Tucker formulation and the implicit Adagrad update, we obtain performances that match state-of-the-art methods on the standard benchmarks and achieve significantly better results for small embedding sizes on several datasets. Compared to the best current algorithm for Tucker decomposition of Balažević et al. (2019b), our approach has less hyperparameters, and we effectively report better performances than the implementation of ComplEx of Lacroix et al. (2018) in the regime of small embedding dimension.

We discuss the related work in the next section. In Section 3, we present a variant of the Tucker decomposition which allows to interpolate between Tucker and CP. The extreme case of this variant, which is equivalent to CP up to a fixed rotation of the embedding, serves as control model to highlight the deficiency of the diagonal rescaling of Adagrad for Tucker-style decompositions in experiments reported in Section 4 . We present the modified version of Adagrad in Section 5 and present experimental results on standard benchmarks of knowledge base completion in Section 7.

## 2   LINK PREDICTION IN KNOWLEDGE BASES

**Notation**   Tensors and matrices are denoted by uppercase letters. For a matrix $U$, $u_i$ is the vector corresponding to the $i$-th row of $U$. The tensor product is written $\otimes$ and the Hadamard product (i.e., elementwise product) is written $\odot$.

### 2.1   LEARNING SETUP

A knowledge base consists of a set $S$ of triples (subject, predicate, object) that represent (true) known facts. The goal of link prediction is to recover facts that are true but not in the database. The data is represented as a tensor $\tilde{X} \in \{0, 1\}^{N \times L \times N}$ for $N$ the number of entities and $L$ the number of predicates. Given a training set of triples, the goal is to provide a ranking of entities for queries of the type (subject, predicate, ?) and (?, predicate, object). Following Lacroix et al. (2018), we use the cross-entropy as a surrogate of the ranking loss. As proposed by Lacroix et al. (2018) and Kazemi & Poole (2018), we include reciprocal predicates: for each predicate $P$ in the original dataset, and given an item $o$, each query of the form (?, $P$, $o$) is reformulated as a query ($o$, $P^{-1}$, ?), where $o$ is now the subject of $P^{-1}$. This doubles the effective number of predicates but reduces the problem to queries of the type (subject, predicate, ?) only.

For a given triple $(i, j, k) \in S$, the training loss function for a tensor $X$ is then

$$\ell_{i,j,k}(X) = -X_{i,j,k} + \log \Big( \sum_{k' \neq k} \exp(X_{i,j,k'}) \Big). \tag{1}$$

For a tensor decomposition model $X(\theta)$ parameterized by $\theta$, the parameters $\hat{\theta}$ are found by minimizing the regularized empirical risk with regularizer $\Lambda$:

$$\hat{\theta} = \operatorname*{argmin}_{\theta} \mathcal{L}(\theta) = \operatorname*{argmin}_{\theta} \frac{1}{|S|} \sum_{(i,j,k) \in S} \ell_{i,j,k}(X(\theta)) + \nu\Lambda(\theta). \tag{2}$$

This work studies specific models for $X(\theta)$, inspired by CP and Tucker decomposition. We discuss the related work on tensor decompositions and link prediction in knowledge bases below.

## 2.2 RELATED WORK

### 2.2.1 CANONICAL DECOMPOSITION AND ITS VARIANTS

The canonical polyadic (CP) decomposition of a tensor $X$ is defined entrywise by

$$\forall i, j, k, \quad X_{i,j,k} = \langle u_i, v_j, w_k \rangle := \sum_{r=1}^{d} u_{ir} v_{jr} w_{kr}.$$

The smallest value of $d$ for which this decomposition exists is the rank of $X$. Each element $X_{i,j,k}$ is thus represented as a multi-linear product of the 3 embeddings in $\mathbb{R}^d$ associated respectively to the $i$th subject, the $j$th predicate and the $k$th object.

CP currently achieves near state-of-the-art performances on standard benchmarks of knowledge base completion (Kazemi & Poole, 2018; Lacroix et al., 2018). Nonetheless, the best reported results are with the ComplEx model (Trouillon et al., 2016), which learns complex-valued embeddings and sets the embeddings of the objects to be the complex conjugate of the embeddings of subjects, i.e., $w_k = \bar{u}_k$. Prior to ComplEx, Dismult was proposed (Yang et al., 2014) as a variant of CP with $w_k = u_k$. While this model obtained good performances (Kadlec et al., 2017), it can only model symmetric relations and does not perform as well as ComplEx. CP-based models are optimized with vanilla Adam or Adagrad and a single regularization parameter (Trouillon et al., 2016; Kadlec et al., 2017; Lacroix et al., 2018) and do not require additional heuristics for training.

### 2.2.2 TUCKER DECOMPOSITION AND ITS VARIANTS

Given a tensor $X$ of size $N \times L \times N$, the Tucker decomposition of $X$ is defined entrywise by

$$\forall i, j, k, \quad X_{i,j,k} = \langle u_i \otimes v_j \otimes w_k, C \rangle := \sum_{r_1=1}^{d_1} \sum_{r_2=1}^{d_2} \sum_{r_3=1}^{d_3} C_{r_1,r_2,r_3} u_{ir_1} v_{jr_2} w_{kr_3}.$$

The triple $(d_1, d_2, d_3)$ are the rank parameters of the decomposition. We also use a multilinear product notation $X = [\![C; U, V, W]\!]$, where $U, V, W$ are the matrices whose rows are respectively $u_j, v_k, w_l$ and $C$ the three dimensional $d_1 \times d_2 \times d_3$ *core tensor*. Note that the CP decomposition is a Tucker decomposition in which $d_1 = d_2 = d_3 = d$ and $C$ is the identity, which we write $[\![U, V, W]\!]$. With a non-trivial core tensor, Tucker decomposition is thus more flexible than CP for fixed embedding size. In knowledge base applications, we typically have $d \leq L \ll N$, so the vast majority of the model parameters are in the embedding matrices of the entities $U$ and $W$. When constraints on the number of model parameters arise (e.g., memory constraints), Tucker models appear as natural candidates to increase the expressivity of the decomposition compared to CP with limited impact on the total number of parameters.

While many variants of the Tucker decomposition have been proposed in the literature on tensor factorization (see e.g., Kolda & Bader, 2009), the first approach based on Tucker for link prediction in knowledge bases is RESCAL (Nickel et al., 2011). RESCAL uses a special form of Tucker decomposition in which the object and subject embeddings are shared, i.e., $U = W$, and it does not compress the relation matrices. In the multilinear product notation above, a RESCAL model is thus written as $X = [\![C; U, I, U]\!]$. Despite some success on a few smaller datasets, RESCAL performances drop on larger datasets (Nickel et al., 2016b). This decrease in performances has been attributed either to improper regularization (Nickel et al., 2011) or optimization issues (Xue et al., 2018). Balažević et al. (2019b) revisits Tucker decomposition in the context of large-scale knowledge bases and resolves some of the optimization and regularization issues using learning rate

annealing, batch-normalization and dropout. It comes at the price of more hyperparameters to tune for each dataset (label smoothing, three different dropouts and a learning rate decay), and as we discuss in our experiments, the results they report are not better than ComplEx for the same number of parameters.

Two methods were previously proposed to interpolate between the expressivity of RESCAL and CP. Xue et al. (2018) expands the HolE model (Nickel et al., 2016b) (and thus the ComplEx model (Hayashi & Shimbo, 2017)) based on cross-correlation of embeddings to close the gap in expressivity with the Tucker decomposition for a fixed embedding size. Jenatton et al. (2012) express the relation matrices in RESCAL as low-rank combination of a family of matrices. We describe the link between these approaches and ours in Appendix 9.4. None of these approach however studied the effect of their formulation on optimization, and reported results inferior to ours.

### 2.2.3 OTHER APPROACHES

**(Graph) neural networks for link prediction**  Several methods have introduced models that go beyond the form of Tucker and canonical decompositions. ConvE (Dettmers et al., 2018) uses a convolution on a 2D tiling of the subject and relation embeddings as input to a 2-layer neural net that produces a new embedding for the pair, then compares to the object embedding. Graph neural networks (Scarselli et al., 2009; Niepert et al., 2016; Li et al., 2016; Bruna et al., 2014) have recently gained popularity and have been applied to link prediction in knowledge bases by Schlichtkrull et al. (2018). This model uses a graph convolutional architecture to generate a variant of CP.

**Poincaré embeddings**  Poincaré embeddings have been proposed as an alternative to usual tensor decomposition approaches to learn smaller embeddings when the relations are hierarchical (Nickel & Kiela, 2017). The method has recently been extended to link prediction in relational data with very good performance trade-offs for small dimensional embeddings on the benchmark using WordNet (Balažević et al., 2019a), which contains relationships such as hypernyms and hyponyms which are purely hierarchical. However, such good results do not extend to other benchmarks.

## 3 INTERPOLATING BETWEEN CP AND TUCKER

In order to better understand the underlying difficulties in learning (variants of) Tucker decompositions compared to CP, our analysis starts from a Tucker model in which the core tensor is itself decomposed with CP. Given a $N \times L \times N$ tensor, a fixed $d$ and assuming a $(d, d, d)$ Tucker decomposition to simplify notation, a Tucker model where the core tensor is itself decomposed with a rank-$D$ CP can be written as (details are given in Appendix 9.3):

$$X_{ijk} = \langle u_i \otimes v_j \otimes w_k, C \rangle = \langle P_1 u_i, P_2 v_j, P_3 w_k \rangle \text{ or equivalently } X = [\![ U P_1^\top, V P_2^\top, W P_3^\top ]\!],$$

where $P_1, P_2, P_3$ are all $D \times d$ matrices. Since most knowledge bases have much fewer predicates than entities ($L \ll N$), the dimension of the predicate factors has little impact on the overall number of model parameters. So in the remainder of the paper, we always consider $P_2 = I$. Learning matrices $U, V, W, P_1, P_3$ of this decomposition simultaneously leads to the following model, which we call CP-Tucker (CPT):

$$(\text{CPT}) \ X_{ijk} = \langle P_1 u_i, v_j, P_3 w_k \rangle, \ u_i, w_k \in \mathbb{R}^d, v_j \in \mathbb{R}^D, P_i \in \mathbb{R}^{D \times d}.$$

The CPT model is similar to a CP model except that the embedding matrices $U$ and $W$ have an additional low-rank constraint ($d$ instead of $D$). We say that the model *interpolates* between CP and Tucker because for $D = d$ it is equivalent to CP (as long as $P_1$ and $P_3$ are full rank), whereas for $D = d^2$ we recover a full Tucker model because the matrices $P_1$ and $P_3$ can be chosen such that $\langle P_1 u_i, v_j, P_3 w_k \rangle = u_i \text{Mat}(v_j) w_k^T$, where $\text{Mat}$ is the operator that maps a $d^2$ vector to a $d \times d$ matrix (see Appendix 9.5).

CPT is similar to CANDELINC (Carroll et al., 1980), except that in CANDELINC the factors $U, V$ and $W$ are fixed and used to compress the data in order to efficiently learn the $P_i$. Closer to CPT, Bro & Andersson (1998) first learn a Tucker3 decomposition of $X$ before applying CANDELINC using the learned factors. These methods are only applicable to least-square estimation, and for tensors of smaller scale than knowledge bases.

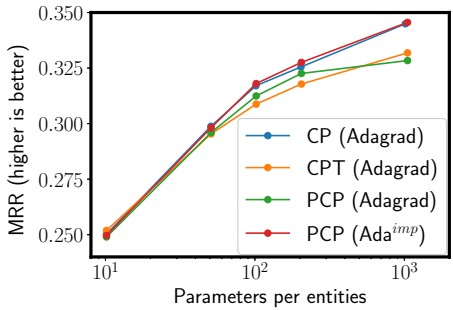 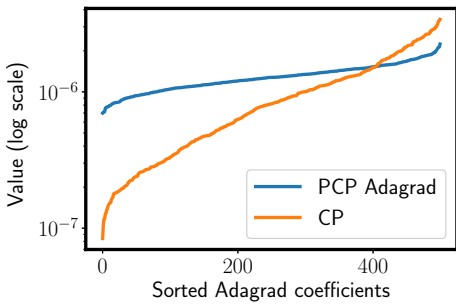

(a) Performances on FB15K-237 in the control experiments for $D = d$, i.e., when PCP is a reparameterization of CP. We observe that PCP and CPT with vanilla Adagrad, which are variants of Tucker, underperform compared to CP. As expected in this case $D = d$, our modification of the Adagrad update leads to the same performances for PCP as for CP.

(b) Adagrad coefficients for subject/object embedding matrices in the control experiments ($D = d$) for CP and PCP, averaged by columns (i.e., embedding dimension) and sorted by values. Adagrad coefficients decay exponentially for CP, but the values are similar across most dimensions in PCP: the fixed unitary transform in PCP removes the benefit of Adagrad.

Figure 1: Results of the control experiment of Section 4.

**Fixed projection matrices: The Projected Canonical Polyadic (PCP) Decomposition** In order to clarify the difficulty that arise when learning a CPT model compared to a CP model, we study a simpler model in which the matrices $P_1$ and $P_3$ are not learned but rather fixed during training and taken as random matrices with orthonormal columns. We call the resulting model the Projected Canonical Polyadic (PCP) decomposition, since $P_1, P_3$ project the embeddings of dimension $d$ into a higher dimension $D$:

$$(\text{PCP}) \ X_{ijk} = \langle P_1 u_i, v_j, P_3 w_k \rangle, \ u_i, w_k \in \mathbb{R}^d, v_j \in \mathbb{R}^D, \ \text{fixed } P_1, P_3 \in \mathbb{R}^{D \times d}$$

When $D = d$ the matrices $P_i$ are then fixed unitary transformations. The PCP (or CPT) model in this case $D = d$ is then equivalent to a CP model, up to a fixed invertible transformation of the embeddings. The capacity of the model grows beyond that of CP as $D$ increases up to $d^2$.

## 4 MOTIVATION: OPTIMIZATION ISSUES WITH CPT AND PCP

As discussed in the related works, previous results suggest that Tucker models are more difficult to train than CP models. The goal of this section is to isolate an issue faced with CPT/PCP models when trained with vanilla adaptive gradient methods such as Adagrad or Adam.

### 4.1 CONTROL EXPERIMENT: UNITARY $P_1$ AND $P_3$ IN PCP

When $D = d$ in PCP, the model becomes equivalent to CP. Indeed, the matrices $P_1$ and $P_3$ are unitary ($P_1 P_1^\top = P_3 P_3^\top = I$) and so $[\![ (U P_1) P_1^\top, V, (W P_3) P_3^\top ]\!] = [\![ U, V, W ]\!]$. There is no practical interest in considering this degenerate case of PCP, we only use it in the following toy experiment to exhibit one of the difficulties encountered when training PCP.

We perform a simple control experiment in which we take one of the standard benchmarks of link prediction in knowledge bases, called FB15K-237, and train a CP model for different values of the rank $D$ and a PCP model with $D = d$ with vanilla Adagrad. The full experimental protocol, including hyperparameter tuning, is similar to our main experiments and is described in Section 7.2. Figure 1a plots the performances in terms of the standard metric mean reciprocal rank (higher is better) as a function of $D$ of CP (blue curve) and PCP (red curve, called PCP (Adagrad)).

We observe that CP obtains significantly better performances than CPT for larger embedding dimension $D$. Since in this toy experiment CP and PCP can represent exactly the same tensors and have equivalent regularizations, the only difference between the algorithms that can explain the difference in performances is in how the optimization is carried out, namely the diagonal rescaling performed by Adagrad: Adagrad adapts the learning rate on a per-parameter basis, depending on previous and

current gradients, and is therefore not invariant by the addition of the matrices $P_1$ and $P_2$ even if these are unitary (we provide the formal justification in the next section). This is shown experimentally in Figure 1b where we plot the average Adagrad coefficients for each embedding dimensions (i.e., adagrad coefficients of subject/object embedding matrices averaged by column). The addition of the random $P_1$ and $P_2$ flattens the Adagrad weights, which in turn removes all the benefit of the adaptive rescaling of the algorithm.

For reference, we also tried to directly learn all parameters including $P_1$ and $P_3$ (i.e., learn a CPT model) with vanilla Adagrad. The performances obtained are also lower than those of CP, as shown in Figure 1a (orange curve).

## 5 A ROTATION INVARIANT ADAGRAD: ADA$^{imp}$

In this section, we study the optimization problem in more details, and more precisely the effect of the diagonal rescaling performed by Adagrad. As a remainder, given a sequence of stochastic gradients $g^{(t)}$ of $\mathcal{L}$ and denoting $G^{(t)} = \epsilon I + \sum_{\tau=1}^{t} g^{(\tau)} g^{(\tau)^\top}$, the (practical) AdaGrad update is:

$$\theta_p^{(t+1)} = \theta_p^{(t)} - \eta g_p^{(t)} / \sqrt{G_{pp}^{(t)}} \quad \text{or equivalently} \quad \theta^{(t+1)} = \theta^{(t)} - \eta \operatorname{Diag}(G^{(t)})^{-1/2}$$

where $\operatorname{Diag}(G)$ is the diagonal matrix obtained by extracting the diagonal elements of $G$.

### 5.1 TWO EQUIVALENT PARAMETRIZATIONS OF PCP

The following decomposition is equivalent to PCP, but its embeddings are expressed in $\mathbb{R}^D$:

$$(\text{PCP}_{\text{FULL}}) \ X_{i,j,k} = \langle P_1 P_1^\top u_i, v_j, P_3 P_3^\top w_k \rangle, \quad \text{with } u_i, v_j, w_k \in \mathbb{R}^D, \text{ fixed } P_1, P_2 \in \mathbb{R}^{D \times d}.$$

Note that with $\underline{u}_i = P_1^\top u_i$ and $\underline{w}_k = P_3^\top w_k$, we go from PCP$_{\text{full}}$ to PCP. The practical differences between PCP and PCP$_{\text{full}}$ are that PCP$_{\text{full}}$ learn embeddings in the high-dimensional space, maintaining the low-rank structure of the overall entity embeddings through the orthogonal projections $P_1 P_1^T$ and $P_3 P_3^T$. The practical interest of PCP$_{\text{full}}$ is not in terms of modeling but rather from the optimization perspective with Adagrad because it has a structure that is closer to that of CP.

Indeed, for $d = D$, $P_1$ and $P_3$ disappear in PCP$_{\text{full}}$, so that optimizing a PCP$_{\text{full}}$ model with Adagrad is equivalent to optimizing a CP model with Adagrad. This property suggests an alternative to the vanilla PCP + Adagrad algorithm, which we call *implicit Adagrad*:

**Implicit Adagrad: Ada$^{imp}$** The approach we propose is to effectively optimize PCP$_{\text{full}}$ with Adagrad. However, when $d < D$, which is the interesting case for PCP, we notice that we do not need to maintain embeddings in $\mathbb{R}^D$. Our approach, called Ada$^{imp}$, computes the gradients and Adagrad coefficients with respect to $u_i, w_k \in \mathbb{R}^D$, but the full dimension factor matrices $U$ and $W$ are never explicitly stored in memory. Rather, we store $\underline{u}_i = P_1^\top u_i$ and $\underline{w}_k = P_3^\top w_k \in \mathbb{R}^d$, which is all that is required for any model computation in PCP$_{\text{full}}$ since $P_1$ and $P_3$ are fixed. Overall, the effective model parameters are exactly the same as in PCP, and we call this approach PCP +Ada$^{imp}$.

An Ada$^{imp}$ update is described in Algorithm 4. While PCP +Adagrad and PCP +Ada$^{imp}$ work with the same number of *model* parameters, the fundamental difference is the computation of Adagrad coefficients. Since Ada$^{imp}$ effectively applies Adagrad to PCP$_{\text{full}}$, we need to maintain the Adagrad coefficients in $\mathbb{R}^D$ even when $d < D$: the overall update is first computed in $\mathbb{R}^D$ and projected back to $\mathbb{R}^d$ *after* the application of Adagrad rescaling. In constrat, in vanilla PCP +Adagrad, the gradient is projected to $\mathbb{R}^d$ *before* Adagrad rescaling.

### 5.2 IMPLICIT OPTIMIZATION OF PCP$_{\text{FULL}}$: ADA$^{imp}$

In this section, we discuss more formally the Ada$^{imp}$ updates, and how they compare to PCP +Adagrad. In the following, $\tilde{U}, \tilde{W}$ are in $\mathbb{R}^{N \times d}$, whereas $U, V$ and $W$ are in $\mathbb{R}^{N \times D}$. Using $d \leq D$, $P_1, P_3 \in \mathbb{R}^{D \times d}$, and we use the notation $\Pi_1 = P_1 P_1^\top$ and $\Pi_3 = P_3 P_3^\top$.

The empirical risk $\mathcal{L}$ can be expressed as a function of three matrices $M^{(1)}$, $M^{(2)}$ and $M^{(3)}$ corresponding to factors of a CP decomposition. We focus on the following problems :

$$(\text{PCP}) \underset{\tilde{U},V,\tilde{W}}{\operatorname{argmin}} \mathcal{L}(\tilde{U}P_1^\top, V, \tilde{W}P_3^\top) \quad (\text{PCP}_{\text{FULL}}) \underset{U,V,W}{\operatorname{argmin}} \mathcal{L}(U\Pi_1^\top, V, W\Pi_3^\top)$$

We focus on a step at time $(t)$ on vectors $\tilde{u}_i$ and $\underline{u}_i$. We assume that at this time $t$, the tensor iterates are the same, that is $\tilde{U} = UP_1^\top$ (resp. $\tilde{W} = WP_1^\top$) so that $[\![\tilde{U}P_1^\top, V, \tilde{W}P_3^\top]\!] = [\![U\Pi_1^\top, V, W\Pi_3^\top]\!]$. In this case, the gradient $\nabla_{M_i^{(1)}}\mathcal{L}$ is the same in both optimization problems, we denote it by $g_i^{(t)}$. Let $G_i^{(t)} = \epsilon I_d + \sum_{\tau=1}^t g_i^{(\tau)} g_i^{(\tau)\top}$. The updates for (PCP) are:

$$\tilde{u}_i^{t+1} = \tilde{u}_i^t - \eta \operatorname{Diag}(P_1^\top G_i^{(t)} P_1)^{-1/2} P_1^\top g_i^{(t)}. \tag{3}$$

Note that due to the presence of $P_1$ inside the Diag operator, the update (3) is not rotation invariant. Moreover, for random $P_1$, the matrix $P_1^\top G_i^{(t)} P_1$ will be far from diagonal with high probability, making its diagonal meaningless. This is visualized in Figure 1b.

Similar updates can be derived for (PCP$_{\text{full}}$):

$$u_i^{t+1} = u_i^t - \eta \operatorname{Diag}(\Pi_1^\top G_i^{(t)} \Pi_1)^{-1/2} \Pi_1^\top g_i^{(t)}. \tag{4}$$

As a sanity check, clearly, for $d = D$ and $\Pi_1 = I$, the update (4) is equivalent to the Adagrad update for the CP model. In the general case $d \leq D$, in order to avoid storing $U \in \mathbb{R}^{N \times D}$, we apply these updates *implicitly* with Ada$^{imp}$, by storing $\underline{u}_i^{(t)} = P_1^\top u_i^{(t)}$ in $\mathbb{R}^d$. Let us compare the updates :

$$
\begin{array}{rlcccc}
 & & & \overbrace{\text{Adagrad in } \mathbb{R}^d} & & \overbrace{\text{projection to } \mathbb{R}^d} \\
(\text{ADAGRAD})\ \tilde{u}_i^{t+1} = \tilde{u}_i^t - \eta & & \times & \operatorname{Diag}(P_1^\top G_i^{(t)} P_1)^{-1/2} & \times & P_1^\top g_i^{(t)} \\
(\text{ADA}^{imp})\ \underline{u}_i^{t+1} = \underline{u}_i^t - \eta & \times\ \underbrace{P_1^\top}_{\text{projection to } \mathbb{R}^d} & \times & \underbrace{\operatorname{Diag}(\Pi_1^\top G_i^{(t)} \Pi_1)^{-1/2}}_{\text{Adagrad in } \mathbb{R}^D} & \times & \underbrace{\Pi_1^\top g_i^{(t)}}_{\substack{\text{projection} \\ \text{to Im}(\Pi) \in \mathbb{R}^D}}
\end{array}
$$

Going back to our control experiment, we note on Figure 1a that PCP +Ada$^{imp}$ matches the performances of CP+Adagrad for all $D$, indicating that we fixed this optimization issue.

## 5.3 ALTERNATIVES TO ADA$^{imp}$

Another solution would be to use Adagrad projected on the column space of $\Pi$, but we show in Appendix 9.1 that even with the diagonal approximation, this is impractical. Note that the version of Adagrad which uses the full matrix $G^{(t)}$ is rotation invariant (see Appendix 9.2 for details), but it cannot be used at the scale of our problems.

It could be argued that the strength of the AdaGrad algorithm in our context mostly comes from its adaptation to the different frequencies of updates of each embedding. In fact, this is one of the examples chosen in Duchi et al. (2011) to display the advantages of AdaGrad compared to stochastic gradient descent. A version of AdaGrad that would only keep one coefficient per embedding (we call this version Ada$^{row}$) would be invariant to unitary transforms by design and would adapt to the various update frequencies of different embeddings. In fact, this version of AdaGrad is used to save memory in Lerer et al. (2019). We test this algorithm in Appendix 9.8. The difference in performances shows that this adaptation is not sufficient to recover the performances of the finer diagonal AdaGrad in our setting. The claim that the approximations made in Ada$^{imp}$ are indeed better is further backed by the experiments in the next section .

## 5.4 COMPLEXITY

The time complexity of our Ada$^{imp}$ update for a batch of size $B$ is $\mathcal{O}(D \cdot d \cdot B)$ which is similar, up to constants, to the complexity of updates for the AdaGrad algorithm. We do not notice any runtime differences between our algorithm applied in dimensions $(d, D)$ and a CP decomposition of dimension $D$ (see Section 7). The runtime for large enough $D$ is dominated by the matrix product ($\mathcal{O}(D^2 \cdot B)$) required to compute the cross-entropy in Equation (1).

---

**Algorithm 1** Step of PComplEx training

$(i, j, k) \leftarrow$ sample from $S$
$g_i, g_j, g_k \leftarrow$ gradients in $(Pu_i, Pu_j, w_k)$
$u_i, \tilde{G}_i \leftarrow \text{Ada}^{imp}(\eta, u_i, g_u, \tilde{G}_i, P)$
$u_j, \tilde{G}_j \leftarrow \text{Ada}^{imp}(\eta, u_j, g_j, \tilde{G}_j, P)$
$w_k, \tilde{G}_k \leftarrow \text{AdaGrad}(\eta, w_k, g_k, G_k)$

---

**Algorithm 2** $\text{Ada}^{imp}$

**Input:** $\eta, x^{(t)}, g^{(t)}, \tilde{G}^{(t-1)}, P$
$\tilde{g}^{(t)} \leftarrow P(P^\top g^{(t)})$
$\tilde{G}^{(t)} \leftarrow \tilde{G}^{(t-1)} + \tilde{g}^{(t)} \odot \tilde{g}^{(t)}$
$x^{(t+1)} \leftarrow x^{(t)} - \eta P^\top \text{Diag}(\tilde{G}^{(t)})^{-1/2} \tilde{g}^{(t)}$
return $x^{(t+1)}, \tilde{G}^{(t)}$

---

Figure 2: The two algorithms used in the training of PComplEx

## 6 PROJECTED COMPLEX

As the state-of-the-art variant of CP is ComplEx (Trouillon et al., 2016; Lacroix et al., 2018), we propose the following alternative to PCP base on ComplEx with $\text{Ada}^{imp}$ in practice. Given the ComplEx decomposition $X = Re([U, V, \overline{U}])$, a low-rank decomposition of the entity factor $U$ as $P\tilde{U}$ leads to the model PComplEx we use in the experiments of Section 7:

$$(\text{PCOMPLEX}) \; X_{ijk} = \langle Pu_i, v_j, \overline{Pu_k} \rangle$$
$$= \langle u_i, P^\top \text{Diag}(v_j)P, \overline{u_k} \rangle, \; u_i, w_k \in \mathbb{C}^d, v_j \in \mathbb{C}^D, \text{ fixed } P \in \mathbb{R}^{D \times d}$$

PComplEx is similar to ComplEx but with interactions described by full matrices of rank $D$ that share a same basis. We learn this decomposition with Algorithms 1 and 4.

## 7 EXPERIMENTS

In this section, we compare ComplEx optimized with AdaGrad and PComplEx optimized with $\text{Ada}^{imp}$. We optimize the regularized empirical risk of Equation (2). Following Lacroix et al. (2018), we regularize ComplEx with the weighted nuclear-3 norm, which is equivalent to regularizing $\|u_i\|_3^3 + \|u_j\|_3^3 + \|w_k\|_3^3$ for each training example $(i, j, k)$. For PComplEx based models, we regularize $\|Pu_i\|_3^3 + \|v_j\|_3^3 + \|Pu_k\|_3^3$ by analogy.

We conduct all experiments on a Quadro GP 100 GPU. The code for PComplEx and $\text{Ada}^{imp}$ is available in the supplementary materials, experiments on ComplEx use the code[1] from (Lacroix et al., 2018). We include results from TuckER (Balažević et al., 2019b), DRT and SRT which are the two models considered in Wang et al. (2019), ConvE (Dettmers et al., 2018), HolEx (Xue et al., 2018), LFM (Jenatton et al., 2012) and MurP (Balažević et al., 2019a) without re-implementation on our parts. All the parameters for the experiments in this section are reported in Appendix 9.11.

### 7.1 DATASETS

WN18 (Bordes et al., 2013) is taken from the Wordnet database which contains words and relations between them. WN18RR (Dettmers et al., 2018) is a filtering of this dataset which removes train/test leakage. YAGO3-10 (Dettmers et al., 2018) is taken from the eponymous knowledge base. Finally, SVO (Jenatton et al., 2012) contains observations of Subject, Verb, Object triples. All statistics for these datasets can be found in Appendix 9.13. Experiments on the FB15K and FB15K-237 datasets are deferred to Appendix 9.10.

### 7.2 RESULTS

We report the filtered Mean Reciprocal Rank (Nickel et al., 2016b) on Figure 3. For SVO, we report the only figure available in previous work which is the filtered hits at $5\%$ (Jenatton et al., 2012). These measures are detailed in Appendix 9.11. Only the grid-search parameters were given for LFM, so we were not able to obtain a precise number of parameters for the number they report.

---

[1] available at `https://github.com/facebookresearch/kbc`

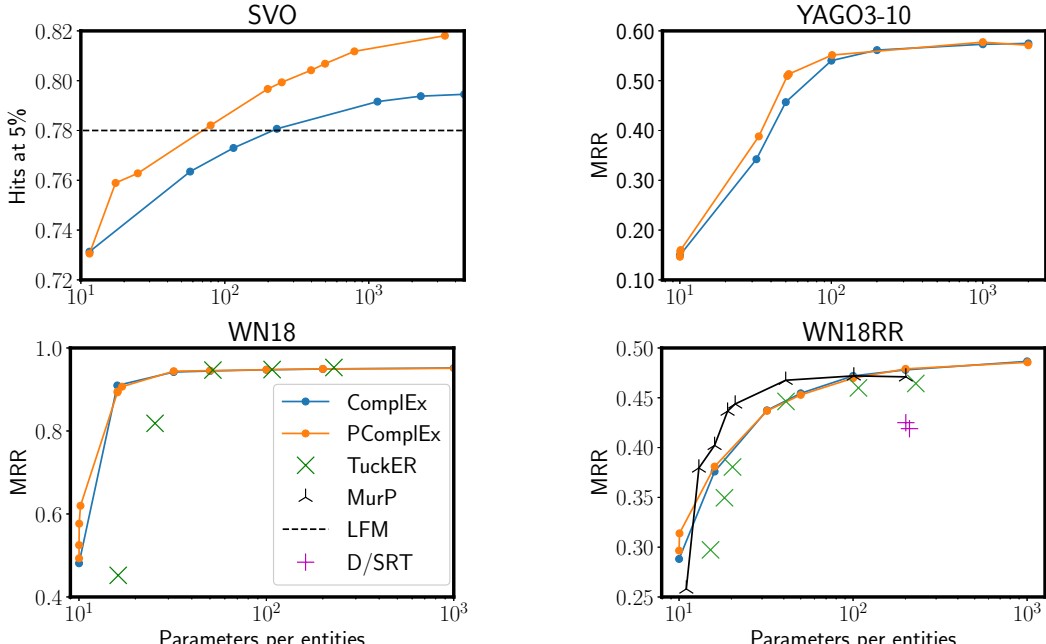

Figure 3: MRR as a function of #floats / entities (see Appendix 9.11) on four knowledge bases. We plot the convex envelope of various operating points we tested, varying $D$ for several values of $d$. For some datasets (WN18, WN18RR), Adam (Kingma & Ba, 2014) is beneficial, in which case we use the implicit adaptation of Adam detailed in Appendix 9.7.

On WN18, SVO and YAGO3-10 we observe sizable performance gains for low embedding sizes : up to $0.14$ MRR points on WN18, $0.05$ MRR points on YAGO and $0.03$ H@5% points on SVO.

The TuckER (Balažević et al., 2019b) model performs similarly to PComplEx and ComplEx except on FB15K and WN18 where it underperforms (see Appendix 9.10). We expect this discrepancy to come from a less extensive grid-search rather than any intrinsic differences in the models that are both based on the Tucker decomposition. The consistency on all operating points of our method with ComplEx shows an advantage of our method, which enjoys the same learning rate robustness as AdaGrad, and does not require choosing a learning-rate decay, leading to easier experiments with only the regularization strength to tune. The MurP model (Balažević et al., 2019a) provides good performances for low embedding sizes on WN18RR, but underperforms on FB15K-237 (see Appendix 9.10). All other models fail to match the performances of ComplEx and PComplEx with equivalent number of parameters.

Variance of performances in PComplEx due to random choice of $P$ is similar to the variance of ComplEx. We present experiments on the WN18 dataset for $5$ different seeds in Appendix 9.9.

## 8 CONCLUSION

By observing that the core tensor of the Tucker decomposition can itself be decomposed, we obtain new models that are reminiscent of the canonical decomposition with low-rank factors. We provide experimental evidence that a naive application of AdaGrad on this decomposition fails, due to individual coordinates losing their meaning. We propose a new algorithm, $\text{Ada}^{imp}$, which fixes this issue. Our model, when optimized with $\text{Ada}^{imp}$, provides better performances than ComplEx in the low-rank regime, and matches its performances in the other regimes.

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

## 9    APPENDIX

In subsections 9.1 and 9.2 where we discuss the Adagrad algorithm, we do so in a general setting, where we study, for fixed $P$ with orthonormal columns, the problem :

$$\min_\theta f(P\theta).$$

### 9.1    PROJECTED ADAGRAD UPDATE

Let $D$ denote $\mathrm{Diag}(G)^{1/2}$ for the classical version of Adagrad and $G^{1/2}$ itself for the "full" version of the update. When the parameter $\theta$ is constrained to a set $\Theta$, the update proposed in Eq.(1) in Duchi et al. (2011) the one obtained by solving

$$\min_{\theta \in \Theta} (\tilde\theta - z)^\top D(\tilde\theta - z) \quad \text{with} \quad z = \tilde\theta^{(t)} - \eta D^{-1} g^{(t)}$$

To enforce the constraint that $\tilde\theta = P\theta$, we can consider the Lagrangian

$$\mathcal{L}(\tilde\theta, \theta; \lambda) = (\tilde\theta - z)^\top D(\tilde\theta - z) - \lambda^\top(\tilde\theta - P\theta)$$

whose stationary points satisfy $D(\tilde\theta - z) = \lambda$  and  $P^\top \lambda = 0$. So this entails $P^\top D(\tilde\theta - \tilde\theta^{(t)}) = \eta P^\top g^{(t)}$ and finally using $\tilde\theta = P\theta$ we obtain an update in $\theta$ as follows

$$\theta^{(t+1)} = \theta^{(t)} - \eta(P^\top DP^\top)^{-1}Pg^{(t)}.$$

Clearly, $PDP^\top$ is in general non-diagonal whether $D$ is diagonal or not, and so this approach does not provide a computationally efficient update.

If $D = G^{1/2}$, then since $PG^{1/2}P^\top = (PGP^\top)^{1/2}$ the update is the same as the full Adagrad update (5) that we derive in the following section and replacing $(PGP^\top)^{1/2}$ by its diagonal approximation recovers update (3).

### 9.2    THE TWO FULL ADAGRAD UPDATES AND THE QUALITY OF APPROXIMATIONS OF THE DIAG VERSIONS

If we consider the full versions of the Adagrad updates then (letting again $\Pi = PP^\top$) its application to $\theta \mapsto f(P\theta)$ and $\tilde\theta \mapsto f(\Pi\tilde\theta)$ yield respectively

$$\tilde\theta^{(t+1)} = \tilde\theta^{(t)} - \eta\, P\big(P^\top G^{(t)}P\big)^{-1/2}P^\top g^{(t)} \quad \text{and} \tag{5}$$

$$\tilde\theta^{(t+1)} = \tilde\theta^{(t)} - \eta\, \Pi\big(\Pi\, G^{(t)}\,\Pi\big)^{-\dagger/2}\Pi\, g^{(t)}, \tag{6}$$

where $M^\dagger$ notes the pseudo-inverse of a matrix $M$. As it turns out, the two updates are equivalent: Indeed, first $(\Pi G\Pi)^{1/2} = P(P^\top GP)^{1/2}P^\top$ because $P^\top P = I$ implies that

$$P(P^\top GP)^{1/2}P^\top P(P^\top GP)^{1/2}P^\top = \Pi G\Pi,$$

and the p.s.d. squareroot is unique. Second, taking the pseudo-inverse of this identity, we have

$$(\Pi G\Pi)^{\dagger/2} = \big(P(P^\top GP)^{1/2}P^\top\big)^\dagger = P(P^\top GP)^{-1/2}P^\top. \tag{7}$$

because, if $H = (P^\top GP)^{1/2}$ is an invertible matrix, then $(PHP^\top)^\dagger = PH^{-1}P^\top$ given that $PHP^\top PH^{-1}P^\top = PP^\top$. Finally multiplying both sides of Eq. (7) by $\Pi$ shows that

$$\Pi(\Pi G^{(t)}\Pi)^{\dagger/2}\Pi = P(P^\top G^{(t)}P)^{-1/2}P^\top.$$

This shows that although Adagrad is not invariant for any invertible $P$, it is invariant for any $P$ such that $P^\top P = I$. Eq.(5) seems in general simpler than (6), but note that if $D = d$, then $\Pi$ is the identity and (6) shows that both full updates are in that case actually equivalent to the full update of Adagrad applied to plain CP.

Finally, the equivalence of the full updates discussed above strongly suggests that if our proposed update performs better than the naive application of Adagrad to PCP, it is because $P\mathrm{Diag}(\Pi G^{(t)}\Pi)^{-1/2}P^\top$ is a better approximation of $(G^{(t)})^{-1/2}$ than $\mathrm{Diag}(PG^{(t)}P^\top)^{-1/2}$ while not being much more computationally expensive.

### 9.3 CP-TUCKER

We have $X = [\![C; U, V, W]\!]$ and $C = [\![P_1, P_2, P_3]\!]$. For all $i, j, k$, we have:

$$
\begin{aligned}
X_{i,j,k} &= \sum_{r_1,r_2,r_3}^{d} C_{r_1,r_2,r_3} U_{i,r_1} V_{i,r_2} W_{k,r_3} \\
&= \sum_{r_1,r_2,r_3}^{d} \left( \sum_{s=0}^{D} [P_1]_{r_1,s} [P_2]_{r_2,s} [P_3]_{r_3,s} \right) U_{i,r_1} V_{i,r_2} W_{k,r_3} \\
&= \sum_{s=0}^{D} \left( \sum_{r_1} U_{i,r_1} [P_1]_{r_1,s} \right) \left( \sum_{r_2} V_{j,r_2} [P_2]_{r_2,s} \right) \left( \sum_{r_3} W_{k,r_3} [P_3]_{r_3,s} \right) \quad = \langle P_1 u_i, P_2 v_j, P_3 w_k \rangle
\end{aligned}
$$

### 9.4 HOLEX AND LATENT FACTOR MODEL

#### 9.4.1 HOLEX

The HolEx model (Xue et al., 2018) writes $X_{i,j,k} = \sum_{r=1}^{R} \langle (c_r \odot u_i) \star u_j, w_k^r \rangle$, where $\star$ denotes the circular correlation[2]. Exploiting the equivalence between HolE and ComplEx (Hayashi & Shimbo, 2017), denoting by $\mathcal{F}$ the discrete Fourier transform (with values in $\mathbb{C}$), we can write for embeddings of size $d$:

$$
\begin{aligned}
\sum_{r=1}^{R} \langle (c_r \odot u_i) \star u_j, w_k^r \rangle &= \frac{1}{d} Re \left( \sum_{r=1}^{R} \langle \overline{\mathcal{F}(c_r \odot u_i)}, \mathcal{F}(u_j), \mathcal{F}(w_k^r) \rangle \right) \\
&= \frac{1}{d} Re \left( \sum_{r=1}^{R} \langle \overline{\mathcal{F}(c_r) \star \mathcal{F}(u_i)}, \mathcal{F}(u_j), \mathcal{F}(w_k^r) \rangle \right)
\end{aligned}
$$

For all vectors, we write $\hat{u}_i = \mathcal{F}(u_i) \in \mathbb{C}^d$. We can re-write the circular correlation $\mathcal{F}(c_r) \star \mathcal{F}(u_i)$ as $C_r \hat{u}_i$ where $C_r \in \mathbb{C}^{d \times d}$ is the circulant matrix associated with $c_r \in \mathbb{R}^d$. We have :

$$
\sum_{r=1}^{R} \langle (c_r \odot u_i) \star v_j, w_k^r \rangle = \frac{1}{d} Re \left( \sum_{r=1}^{R} \langle C_r \overline{\hat{u}_i}, \hat{u}_j, \hat{w}_k^r \rangle \right).
$$

Finally, with $C^1 = [C_1, \ldots, C_R] \in \mathbb{R}^{Rd \times d}$ the vertical stacking of all $C_r$, $C^2 = [I_d, \ldots, I_d]$ and $\hat{w}_k = [\hat{w}_k^1, \ldots, \hat{w}_k^R]$:

$$
\sum_{r=1}^{R} \langle (c_r \odot u_i) \star u_j, w_k^r \rangle = \frac{1}{d} Re \left( \langle C^1 \overline{\hat{u}_i}, C^2 \hat{u}_j, \hat{w}_k \rangle \right)
$$

HolEx with embeddings of size $d$ is close to the CPT with $D = Rd$, allowing for two different complex matrices to act on left and right hand side embeddings.

#### 9.4.2 LATENT FACTOR MODEL

The latent factor model defines the score for a triple $(i, j, k)$ as:

$$
X_{i,j,k} = \langle (s_i + z), R_j(o_k + z') \rangle, \quad \text{with } R_j = \sum_{r=1}^{D} \alpha_r^j u_r v_r^\top.
$$

Removing the bias terms $z$ and $z'$ and gathering $u_r$ and $v_r$ into matrices $P_1$ and $P_3$ leads to the model CPT. In the PCP model, we fix $P_1$ and $P_3$ instead of learning them. We do not use a sparsity inducing penalty on $\alpha$ but rather a variational form of the nuclear norm on the whole tensor.

---

[2] $[a \star b]_k = \sum_{i=0}^{d-1} a_i b_{(i+k) \bmod d}$.

## 9.5 TUCKER2 WITH CP-TUCKER

Let for all $1 \leq r \leq d$, $M^{(r)}$ be a matrix of zeros except its $r - th$ column which is all one. Let $P_1$ be the vertical concatenation of all $(M^{(r)})_{r=1..d}$ and $P_2$ the vertical concatenation of $d$ identity matrix in $\mathbb{R}^d$. Remember that for all $k$, $w_k$ is an element of $\mathbb{R}^{d^2}$. For all $0 \leq r < d$, let $w_k^r$ be the restriction of $w_k$ to its $[rd, (r+1)d]$ coordinates.

Then $P_1$ and $P_2$ are elements of $\mathbb{R}^{d^2 \times d}$ and we have for all $i, j, k$:

$$\langle P_1 u_i, P_2 v_j, w_k \rangle = \sum_{r_1=1}^{d} \langle M^{(r_1)} u_i, I_d v_j, w_k^{r_1} \rangle$$

$$= \sum_{r_1=1}^{d} u_{i,r_1} \langle v_j, w_k^{r_1} \rangle \quad \text{by definition of } M^{(r_1)}$$

$$= \sum_{r_1=1}^{d} u_{i,r_1} \left( \sum_{r_2=1}^{d} v_{j,r_2} w_{k,r_1 r_2} \right) \quad \text{by definition of } w_k^{r_1}$$

$$= u_i^\top \text{Mat}(w_k) v_j$$

## 9.6 COMPLETE PCOMPLEX ALGORITHM

Let $U \in \mathbb{C}^{N \times d}$ be the the entity embedding matrix and $V \in \mathbb{C}^{2L \times D}$ be the predicate embedding matrix. Let $\mathcal{P} : \mathbb{R}^{N \times d} \mapsto \mathbb{R}^{N \times D}$ such that $\mathcal{P}(U_i) = P U_i$. Let $G_t^U$ and $G_t^V$ be the stochastic gradients with respect to $U$ and $V$ at time $t$.

---

**Algorithm 3** PComplEx optimized with Ada$^{imp}$

---

**Input:** learning rate $\eta$, (random) matrix $P$ with orthogonal columns, $\epsilon$
  $C_U, C_V \leftarrow 0$
  **while** $U, V$ not converged **do**
    $\tilde{G}_t^U \leftarrow \mathcal{P}(G_t^U)$
    $C_U \leftarrow C_U + \tilde{G}_t^U \odot \tilde{G}_t^U$
    $C_V \leftarrow C_V + G_t^V \odot G_t^V$
    $U \leftarrow U - \eta \cdot \mathcal{P}^\top (\tilde{G}_t^U / (\sqrt{C_U + \epsilon}))$
    $V \leftarrow V - \eta \cdot G_t^V / (\sqrt{C_V + \epsilon}))$
  **end while**
  return $U$

---

## 9.7 ADAM - IMPLICIT

For WN18RR, we used the ideas presented in this paper, but adapted to the Adam (Kingma & Ba, 2014) optimization algorithm. Similar to Ada$^{imp}$, first and second moment estimates are accumulated in $\mathbb{R}^D$ and we project back to $\mathbb{R}^d$ only for the parameter update. For simplicity, we present here the dense version of the algorithm applied to the entity embeddings $U \in \mathbb{R}^{N \times d}$. Let $\mathcal{P} : \mathbb{R}^{N \times d} \mapsto \mathbb{R}^{N \times D}$ such that $\mathcal{P}(U_i) = P U_i$. Let $G_t$ be the stochastic gradient with respect to $U$ at time $t$.

## 9.8 ADA$^{row}$

For the same control experiment as in Section 5, we observe the performances of Ada$^{row}$ which is rotation invariant by design.

---

**Algorithm 4** $\text{Adam}^{imp}$ applied to $U$

---

**Input:** $\eta, \beta_1, \beta_2, \mathcal{P}, \epsilon$
  $m_0, v_0, t \leftarrow 0$
  **while** $U$ not converged **do**
    $t \leftarrow t + 1$
    $\tilde{G}_t \leftarrow \mathcal{P}(G_t)$
    $m_t \leftarrow \beta_1 \cdot m_{t-1} + (1 - \beta_1) \cdot \tilde{G}_t$
    $v_t \leftarrow \beta_2 \cdot v_{t-1} + (1 - \beta_2) \cdot \tilde{G}_t \odot \tilde{G}_t$
    $\hat{m}_t \leftarrow m_t / (1 - \beta_1^t)$
    $\hat{v}_t \leftarrow v_t / (1 - \beta_2^t)$
    $U \leftarrow U - \eta \cdot \mathcal{P}^\top (\hat{m}_t / (\sqrt{\hat{v}_t} + \epsilon))$
  **end while**
  return $U$

---

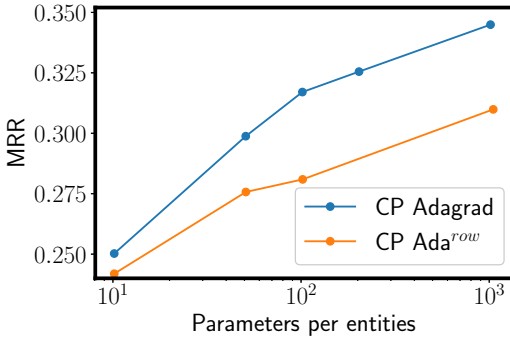

Figure 4: $\text{Ada}^{row}$ vs AdaGrad on FB15K-237.

## 9.9 VARIANCE OF PCOMPLEX

We run 5 grid search and plot the 5 associated convex-hulls on the WN18 dataset optimized with $\text{Ada}^{imp}$. Note that despite the added randomness of the choice of $P$, there is not more variance in PComplEx than in ComplEx.

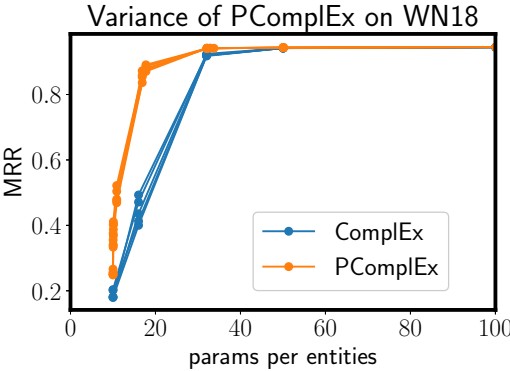

## 9.10 FB15K DATASETS

We use two subsets of the Freebase knowledge base : FB15K (Bordes et al., 2013) and FB15K-237 (Toutanova & Chen, 2015). FB15K-237 is a harder version of FB15K, where some triples have been removed to avoid leakage between the train and test set. There is no difference of performances between PComplEx and ComplEx on these datasets.

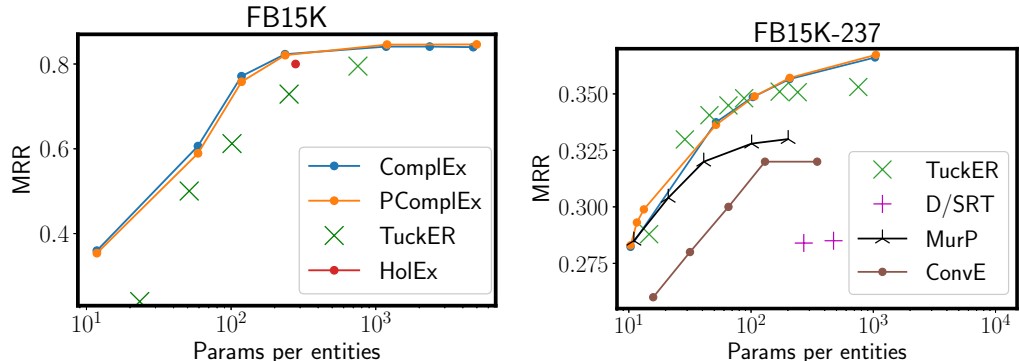

## 9.11 EXPERIMENTAL DETAILS

**Metrics**  Let $rank(\hat{X}_{i,j,:};k)$ be the rank of $\hat{X}_{i,j,k}$ in the sorted list of values of $\hat{X}_{i,j,:}$. We report the MRR for most datasets :

$$MRR(X) = \frac{1}{|S|} \sum_{(i,j,k)\in S} \frac{1}{rank(X_{i,j,:};k)}.$$

For SVO, the task is slightly different as the ranking happens on the predicate mode. The metric reported in Jenatton et al. (2012) is Hits@5% defined as :

$$H@5\%(X) = \frac{1}{|S|} \sum_{(i,j,k)\in S} \mathbb{1}(rank(X_{i,:,k};j) \leq 227).$$

The metrics we report are filtered by ignoring other true positives when computing the ranks, as done in Bordes et al. (2013).

**Number of parameters per entities**  We count the number of floats in the model, and divide by the number of entities in the dataset. For different methods, the number of parameters are :

- ComplEx: $2 \cdot d \cdot (N + 2L)$
- PComplEx: $2 \cdot d \cdot N + 2 \cdot D \cdot 2L + d \cdot D$
- TuckEr: $N \cdot d_e + L \cdot d_p + d_e^2 \cdot d_p$
- MurP: $N \cdot (d + 1) + 2 \cdot L \cdot d$
- ConvE: taken from Dettmers et al. (2018)
- D/SRT: taken from Wang et al. (2019)
- HolEx: $d \cdot (N + L)$

**Grid Search**  For SVO:

- For Complex vary $d$ in $[5, 25, 50, 100, 500, 1000, 2000]$. For PComplEx, we vary $d$ in $[5, 25, 50, 100, 500]$.
- The strength of the regularizer, $\nu$ varies in $[5e - 4, 1e - 3, 5e - 3, 1e - 2, 5e - 2, 1e - 1]$.
- Finally, for PComplEx, we vary the dimension $D$ in $[5, 25, 50, 100, 500, 1000, 2000, 4000, 8000]$.

For all other datasets, we run 500 epochs :

- For FB15K and FB15K-237, we vary $d$ in $[5, 25, 50, 100, 500, 1000, 2000]$. For YAGO3-10, WN18 and WN18RR, we add ranks 8 and 16 to that list.
- The strength of the regularizer, $\nu$ varies in $[5e - 4, 1e - 3, 5e - 3, 1e - 2, 5e - 2, 1e - 1]$.
- Finally, for PComplEx, we vary the dimension $D$ in $[5, 25, 50, 100, 500, 1000, 2000]$.

For TuckEr, we use the hyperparameters described in Balažević et al. (2019b) for each dataset. We apply a multiplicative factor $\gamma$ to the dropout rates and run a grid-search over this parameter.

- For WN18RR, we vary $d_e$ in $[12, 15, 18, 20, 40, 100, 200]$. For FB15K-237, we vary $d_e$ in $[10, 20, 30, 40, 50, 80, 100, 200]$. For WN18, $d_e$ varies in $[5, 8, 16, 25, 50, 100]$. For FB15K $d_e$ varies in .
- For WN18RR, WN18 we search $\gamma$ over $[0, 0.5, 1]$. For FB15K-237 where this lead to non increasing performances as a function of number of parameters, we refined this grid to $[0, 0.1, 0.2, ..., 0.7]$ for $d_e \leq 80$. For FB15K, we use a grid of $[0, 0.2, 0.4, ...1]$.

For MurP, we use the hyperparameters described in Balažević et al. (2019a) for each dataset and vary $d$ in $[10, 12, 15, 18, 20, 40, 100, 200]$ on WN18RR and in $[10, 20, 40, 100, 200]$ for FB15K-237.

**Other details** Batch-size is fixed to 1000, the learning-rate is $1e - 1$ both for Adagrad and for $\text{Ada}^{imp}$, we run 100 epochs to ensure convergence. The best model based on validation MRR is selected and we report the corresponding test MRR.

### 9.12 RUNNING TIMES

We give here the running times of each method per epoch, on the WN18RR dataset for comparable dimensionalities and run on a P100 GPU. We use the original implementation for MurP (Balažević et al., 2019a) and TuckEr (Balažević et al., 2019b). For ComplEx, we use the implementation from Lacroix et al. (2018).

- ComplEx ($d = 200$) : 4s/epoch.
- PComplEx ($d = D = 200$, $\text{Ada}^{imp}$) : 5s/epoch.
- MurP ($d = 200$): 38s/epoch.
- TuckEr ($d_e = 200$, $d_r = 200$): 24s/epoch

Note that these running times are implementation dependent. In the figure below, we give the learning-curves of MurP, TuckEr, ComplEx and PComplEx for one operating point on the WN18RR dataset. The convergence speed of these methods is given in the figure below at an operating point on WN18RR where all methods are close in final performances.

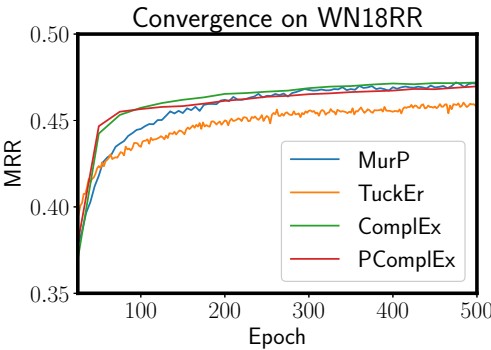

## 9.13 DATASET STATISTICS

| Dataset | N | L | Train |
|---------|------|------|-------|
| FB15K | 15k | 1k | 500k |
| FB15K-237 | 15k | 237 | 272k |
| WN18 | 41k | 18 | 141k |
| WN18 | 41k | 11 | 141k |
| YAGO3-10 | 123k | 37 | 1M |
| SVO | 30k | 4.5k | 1M |

Table 1: Dataset statistics.

