# OpenReview forum: "Projected Canonical Decomposition for Knowledge Base Completion"
_ICLR.cc/2020/Conference — Reject_

### Official Review · AnonReviewer2 · 2019-10-23
**Official Blind Review #2**

**Rating:** 3

**Review:**

In this paper, a tensor decomposition method is studied for link prediction problems. The model is based on Tucker decomposition but the core tensor is decomposed as CP decomposition so that it can be seen as an interpolation between Tucker and CP. The performance is evaluated with several NLP data sets (e.g., subject-verb-object triplets).

Although the entire idea is interesting, the current form of the paper is not sufficient for acceptance. The main reasons are (A) the proposed model is not completely novel and (B) the empirical results are not significant.

(A) The idea of combining CP and Tucker is not new. For example, Tomioka et al. (2010; Section 3.4) considered the Tucker-CP patterns (CP decomposition of the Tucker core). Although they used the Tucker-CP model to improve the interpretability rather than link prediction, the paper needs to make some attribution to the prior work.

(B) By looking Figure 3, the proposed method, PComplEx, is not significantly better than the existing methods such as ComplEx. Except SVO data, PComplEx and ComplEx share almost the same performance curve. Also, other existing methods such as TuckER and MurP are evaluated only in a few points while (P)ComplEx is evaluated in many points. I feel this is unfair.

Tomioka, R., Hayashi, K., & Kashima, H. (2010). Estimation of low-rank tensors via convex optimization. arXiv preprint arXiv:1010.0789.

**Experience Assessment:**

I have published in this field for several years.

**Review Assessment: Checking Correctness Of Derivations And Theory:**

I assessed the sensibility of the derivations and theory.

**Review Assessment: Checking Correctness Of Experiments:**

I assessed the sensibility of the experiments.

**Review Assessment: Thoroughness In Paper Reading:**

I made a quick assessment of this paper.

---

> ### Author Response · Authors · 2019-11-12
> **detailed comments**
>
> (A) The idea of combining CP and Tucker is not new. For example, Tomioka et al. (2010; Section 3.4) considered the Tucker-CP patterns (CP decomposition of the Tucker core). Although they used the Tucker-CP model to improve the interpretability rather than link prediction, the paper needs to make some attribution to the prior work.
> → Indeed, the idea of combining CP and Tucker is far from new (we cite CANDELINC from 1980 and a method from Bro & Andersson from 1998). The interest of this paper is the method for optimization which differs from all of these prior work (and the work from Tomioka, Hayashi & Kashima) due to the tasks and scales considered. On these datasets, the loss is no longer Frobenius and the use of adaptive stochastic method is critical to obtain state of the art results. We show that adaptive algorithms are crucial to learn these decompositions in this context and that the diagonal approximation made in practical implementations of these algorithms is too crude to learn the Tucker decomposition.
>
> (B) By looking Figure 3, the proposed method, PComplEx, is not significantly better than the existing methods such as ComplEx. Except SVO data, PComplEx and ComplEx share almost the same performance curve. Also, other existing methods such as TuckER and MurP are evaluated only in a few points while (P)ComplEx is evaluated in many points. I feel this is unfair.
> → Regarding the evaluation of other methods, please, see the general comment. For the gain in performance: note that we provide curves whereas the standard in the field is tables for a fixed number of parameters. The maximal gains we observe for a fixed number of parameters are substantial on other datasets: +0.14 MRR (absolute) on WN18 and +0.05 MRR on YAGO. For SVO, our method provides better performances on all operating points compared to ComplEx, and by a fair margin.

---

### Official Review · AnonReviewer3 · 2019-10-25
**Official Blind Review #3**

**Rating:** 8

**Review:**

The authors present a new way of decomposing 3-order tensors which uses interpolation
between the Tucker and CP decompositions, called CPT. The main idea is to present the components of the CP model
with an additional low-rank structure.
The authors also provide a new optimization algorithm called ADA-imp, for learning this decomposition,
which is a variant of Adagrad adapted to their settings.
The paper is overall interesting, clearly written and well-motivated.
The mathematical derivations are, as far as I could follow, correct and non-trivial.  (I did not read all the details in the Appendix).
The authors also show favorable experimental results on two knowledge-base datasets, with improved loss vs. #parameter used tradeoff.
A few unclear issues and suggestions for improvements are below.

The authors present the problem as completion of a binary 3-order tensor, i.e. predicting for triplets (subject, predicate, ?) if '?' refers to 0 or 1.
But they also write 'we formulate this problem as a multi-class classification problem, where the classes are the entities of the knowledge base'  - so this is not a binary problem? does this mean there is some structure that must be present in the tensor? (e.g. there is exactly one '1' in each column of length N? This should be clarified.

It would be good to make the description of Algorithms 1 and 2 more precise and detailed.
For example, the operation/algorithm AdaGrad(\eta;w_k; g_k;G_k) is not defined. AdaGrad is described in the Appendix but it is hard to match it to get the precise operation used in Algorithm 1.
Algorithm 1 shows one step of PComplEx, and it would be good to add the entire PComplEx algorithm, with input,output&parameters.

The authors present their method in the context of knowledge base completion, thus for tensors of order 3, but it is not clear if any of the components they proposed indeed specialized for this problem, or is it a contribution to general tensor decomposition. Some remarks regarding the (in?)applicability of the method more generally would be helpful.

Figure 3 describing the experimental results should be explained better. There are few methods shown only in some of the graphs and only for some parameter values - why?
The complexity measure 'parameters-per-entity' should be clearly defined (I didn't find it in the text). Similarly, the performance measures 'mean reciprocal rank' and 'hits at 5%'
should be defined in terms of the tensor.
The authors should also add running times of the different experiments and methods.


Minor:
--------
In the main paper, the authors define an (N,L,N) tensor, but in the appendix Section 9.9 they list N and P. Does P refer to L here?

The authors mention a few times usage of 'deep-learning techniques' - but I believe that in at least some of the contexts, they refer to optimization methods which are typically used in deep learning, and  are applied here to train other models presented in the text, and not to the usage of actual deep learning architectures - this is confusing and should be clarified.

Page 7, top: what are the matrices M^(1), M^(2), M^(3)? they seem to be different for different decompositions



**Experience Assessment:**

I have read many papers in this area.

**Review Assessment: Checking Correctness Of Derivations And Theory:**

I assessed the sensibility of the derivations and theory.

**Review Assessment: Checking Correctness Of Experiments:**

I carefully checked the experiments.

**Review Assessment: Thoroughness In Paper Reading:**

I read the paper at least twice and used my best judgement in assessing the paper.

---

> ### Author Response · Authors · 2019-11-12
> **detailed comments**
>
> The authors present the problem as completion of a binary 3-order tensor, i.e. predicting for triplets (subject, predicate, ?) if '?' refers to 0 or 1. But they also write 'we formulate this problem as a multi-class classification problem, where the classes are the entities of the knowledge base' - so this is not a binary problem? does this mean there is some structure that must be present in the tensor? (e.g. there is exactly one '1' in each column of length N? This should be clarified.
> → Despite the ground truth tensor being binary, the evaluation of choice in this field is done by ranking. Hence, the estimate we learn is a tensor of scores for each triple (subject, predicate, object). This does not assume any particular structure on the columns (mode-3 fibers in our case). We use the cross-entropy as a surrogate for the ranking loss : if there are several ones in a fiber of the ground truth tensor our model should learn a uniform distribution over these objects.
>
> It would be good to make the description of Algorithms 1 and 2 more precise and detailed. For example, the operation/algorithm AdaGrad(\eta;w_k; g_k;G_k) is not defined. AdaGrad is described in the Appendix but it is hard to match it to get the precise operation used in Algorithm 1. Algorithm 1 shows one step of PComplEx, and it would be good to add the entire PComplEx algorithm, with input, output & parameters.
> → We added the full algorithm in the supplementary materials. (Appendix 9.6).
>
> The authors present their method in the context of knowledge base completion, thus for tensors of order 3, but it is not clear if any of the components they proposed indeed specialized for this problem, or is it a contribution to general tensor decomposition. Some remarks regarding the (in?)applicability of the method more generally would be helpful.
> → No component of ADA^imp is specialized to tensors of order 3 and could be readily re-used for tensors of higher order. We present it here for the order 3 due to the application we target, for which adaptive algorithms (Adagrad / Adam) seems to be critical.
>
> Figure 3 describing the experimental results should be explained better. There are few methods shown only in some of the graphs and only for some parameter values - why?
> → This issue is addressed in the general comment.
>
> The complexity measure 'parameters-per-entity' should be clearly defined (I didn't find it in the text).
> → Parameters per entity are the total amount of parameters divided by the total number of entities. Precise formulas for each method has been added in the supplementary (Appendix 9.11).
>
> Similarly, the performance measures 'mean reciprocal rank' and 'hits at 5%' should be defined in terms of the tensor.
> → We added the precise definition of these metrics in the supplementary materials. (Appendix 9.11)
>
> The authors should also add running times of the different experiments and methods.
> → Running times as well as a convergence curve have been added in the supplementary materials (Appendix 9.12).
>
>  Minor: In the main paper, the authors define an (N,L,N) tensor, but in the appendix Section 9.9 they list N and P. Does P refer to L here?
> → yes sorry. This is fixed in the revision
>
> The authors mention a few times usage of 'deep-learning techniques' - but I believe that in at least some of the contexts, they refer to optimization methods which are typically used in deep learning, and are applied here to train other models presented in the text, and not to the usage of actual deep learning architectures - this is confusing and should be clarified.
> →Deep learning techniques here refer specifically to dropout, batch-normalization and learning rate annealing. This is clarified in the revision.
>
> Page 7, top: what are the matrices M^(1), M^(2), M^(3)? they seem to be different for different decompositions
> → Indeed. M^(1) is UP_1 for PCP, but U for CP or UPi_1 for PCP_full.
> Since all these methods compute their final score in a CP fashion, we study the gradient with respect to the CP "factors" which are computed differently for different methods.

---

### Official Review · AnonReviewer1 · 2019-10-28
**Official Blind Review #1**

**Rating:** 3

**Review:**

* Summary:
The paper introduces a novel tensor decomposition that is reminiscent of canonical decomposition (CP) with low-rank factors, based on the observation that the core tensor in Tucker decomposition can be decomposed, resulting in a model interpolating between CP and Tucker. The authors argue that a straight application of AdaGrad on this decomposition is inadequate, and propose Ada^{imp} algorithm that enforces rotation invariance of the gradient update. The new decomposition is applied to ComplEx model (called PComplEx) that demonstrates better performance than the baseline.

* Comments:
Although the approach is well motivated, the paper has many ambiguities that need to better clarification.
1. Tucker decomposition results in lower dimension factors, "d" in the paper. So the resulting core tensor is of size (d \times d \times d). However, this core tensor is further decomposed with a rank-D CP as shown in Section 3, where D >= d. Basically, first the original tensor is factored into lower rank d, and the core tensor is then expanded into rank D >= d. The reader did not understand what is the justification for this approach? Please provide further explanation on this part.
2. The confusion of P_2 and P_3 terms in the paper. At the beginning of Section 3, P_2 is assumed to be identity through out the paper. But P_2 is mentioned to have specific attributes in other parts of the paper, such as in the second paragraph from the bottom of page 4, the first paragraph and first equation on page 5. And P_2 does not appear in AdaGrad algorithm.
3. The experiment is lacking. First, the paper does not explain the meaning of evaluation metrics. Second, the authors do not provide an insight, why PComplEx is better than the ComplEx baseline on SVO dataset, but performs similarly on other datasets. Which factors lead to such improvement?
4. The comparison to other state-of-the-arts is inadequate, each compared method only has one or few configurations in terms of number of parameters.

Overall the proposed decomposition method might have significant contribution to research progress in this field, but the paper fails to convince the reader of its significance. I feel the paper should be overhauled.

**Experience Assessment:**

I have read many papers in this area.

**Review Assessment: Checking Correctness Of Derivations And Theory:**

I assessed the sensibility of the derivations and theory.

**Review Assessment: Checking Correctness Of Experiments:**

I assessed the sensibility of the experiments.

**Review Assessment: Thoroughness In Paper Reading:**

I read the paper at least twice and used my best judgement in assessing the paper.

---

> ### Author Response · Authors · 2019-11-12
> **detailed comments**
>
> 1. Tucker decomposition results in lower dimension factors, "d" in the paper. So the resulting core tensor is of size (d \times d \times d). However, this core tensor is further decomposed with a rank-D CP as shown in Section 3, where D >= d. Basically, first the original tensor is factored into lower rank d, and the core tensor is then expanded into rank D >= d. The reader did not understand what is the justification for this approach? Please provide further explanation on this part.
>
> → We start from a tensor of size n x n x p. A Tucker decomposition of rank d leads to :
> d x (n +n + p) parameters for the factors and d x d x d parameters for the core tensor.
> In order to link this decomposition with the CP decomposition which is easier to optimize, we further decompose this core tensor with a CP decomposition of rank D. Thus, d x d x d parameters become d x (D + D + D) (which is smaller than d x d x d as long as D < d^2/3).
> We allow D > d because a tensor of shape d x d x d can have a CP rank as high as d^2.
>
>
> 2. The confusion of P_2 and P_3 terms in the paper. At the beginning of Section 3, P_2 is assumed to be identity through out the paper. But P_2 is mentioned to have specific attributes in other parts of the paper, such as in the second paragraph from the bottom of page 4, the first paragraph and first equation on page 5. And P_2 does not appear in AdaGrad algorithm.
> → There is indeed a confusion between P_2 and P_3 in the paper,  we thank the reviewer for pointing this out. Since P_2 is assumed to be the identity, it should not appear in the paper outside of the definition of CPT (beginning of Section 3). All further occurrences of P_2 are typos and have been fixed in the revision.
>
> 3. The experiment is lacking. First, the paper does not explain the meaning of evaluation metrics. Second, the authors do not provide an insight, why PComplEx is better than the ComplEx baseline on SVO dataset, but performs similarly on other datasets. Which factors lead to such improvement?
> → Regarding evaluation metrics, we have added the definition of the mean reciprocal rank and hits@5% in Appendix 9.11. We attribute the difference in performance on SVO to a difference in the underlying structure of the data that makes Tucker decomposition particularly suited. Similarly to MurP being better on WN18RR than on FB237, it is possible that SVO is a dataset that is more amenable to a Tucker decomposition.
>
> 4.The comparison to other state-of-the-arts is inadequate, each compared method only has one or few configurations in terms of number of parameters.
> → We performed new experiments. Please, see the general comments.

---

### Author Response · Authors · 2019-11-12
**Additional experiments for Tucker and MurP**

We thank all reviewers for their comments. We address more general issues here, and answer more particular points in separate comments. One of the main criticism of the reviewers is that methods from the state-of-art other than Complex are not evaluated on all operating points in terms of rank.
We initially reported for each algorithms, the performances reported by the authors of each method. Considering the reviewers concerns, we have re-run MurP [1] and TuckEr [2] (we chose those because their code is publicly available, and they were close in performances to our methods). We updated Figure 3 and Appendix 9.10 by adding a complete rank profile on WN18RR, FB237 for TuckEr and MurP and also on WN18 and FB15K (*) for TuckEr (see Appendix 9.11 for a detailed description of the experimental protocol). With more operating points  for these algorithms :

* It is confirmed that TuckEr performs essentially similarly to PComplEx on most operating points for WN18RR and FB237. The small differences in performance can most likely be explained by the difference in loss / label smoothing used in the two set-ups. However, our model is much simpler to tune, as it only has one regularization parameter, and its optimization procedure is well understood as shown in our work. TuckEr underperforms on WN18 and FB15k.
* MurP performs better than PComplEx and TuckEr on some operating points of WN18RR but severely underperforms on FB237 as the dimensionality increases.
* Neither Tucker nor MurP matches the performances of ComplEx for higher dimensionalities, in contrast to PComplEx which, by design, is equivalent to Complex for d=D.

In conclusion, PComplEx optimized with AdaImp (or AdamImp for WN datasets) has less hyperparameters, and has performances that do not deteriorate at high ranks, while matching TuckEr's performances at lower parameters per entities. It also leads to faster convergence as shown in Appendix 9.12.

[1] Ivana Balazevic, Carl Allen and Timothy Hospedales. Multi-relational poincaré graph embeddings. ArXiv 2019
[2] Ivana Balazevic, Carl Allen and Timothy Hospedales. TuckEr: Tensor factorization for knowledge graph completion. EMNLP 209

(*) We did not run TuckEr and MurP on SVO because this would require coding a new forward pass for the model and tune all 6 hyperparameters of the method from scratch (because SVO’s task is to answer queries of the form (subject, ?, object)). We also did not run these models on YAGO since an epoch takes 335s with the available implementation leading to 10h experiments for 100 epochs making the tuning of all hyperparameters impractical. Nonetheless, we believe the current experiments are sufficient to support the conclusions in the paper.

---

### Decision · Program_Chairs · 2019-12-19

**Decision:**

Reject

**Comment:**

The paper proposes a tensor decomposition method that interpolates between Tucker and CP decompositions. The authors also propose an optimization algorithms (AdaImp) and argue that it has superior performance against AdaGrad in this tesnor decomposition task. The approach is evaluated on some NLP tasks.
The reviewers raised some concerns related to clarity, novelty, and strength of experiments. As part of addressing reviewers concerns, the authors reported their own results on MurP and Tucker (instead of quoting results from reference papers). While the reviewers greatly appreciated these experiments as well as authors' response to their questions and feedback, the concerns largely remained unresolved. In particular, R2 found the gain achieved by AdaImp not significantly large compared to Adagrad. In addition, R2 found very limited evaluation on how AdaImp outperforms Adagrad (thus little evidence to support that claim). Finally, AdaImp lacks any theoretical analysis (unlike Adagrad).